# Sex Differences in the Prevalence of Chronic Pain in Mid-Life: A Systematic Review and Meta-Analysis

**DOI:** 10.3390/biomedicines13102523

**Published:** 2025-10-16

**Authors:** Catherine Borra, Jessica Pawson, Nathalie Rich, Rebecca Hardy

**Affiliations:** 1UCL Social Research Institute, Institute of Education, University College London, London WC1H 0NU, UK; 2BARTS Bone & Joint Health, Blizard Institute, Faculty of Medicine and Dentistry, Queen Mary’s University London, London E1 2AT, UK; 3Division of Psychiatry, Faculty of Brain Sciences, University College London, London WC1N 3AR, UK; 4School of Sport, Exercise and Health-Loughborough University, Loughborough LE11 3TU, UK

**Keywords:** chronic pain, chronic regional pain, chronic widespread pain, sex differences, gender, pain prevalence

## Abstract

**Background/Objectives**: Chronic pain (CP) affects more females than males, but it is unclear how differences present at mid-life, a period characterized by distinct changes which may exacerbate inequality. **Methods**: Using a search strategy combining MeSH terms and Boolean operators, we searched MEDLINE, EMBASE, AMED, and PSYCHinfo for population-representative cohort or cross-sectional studies of CP prevalence. We conducted a systematic review of CP prevalence by sex and the difference in prevalence of CP between sexes at mid-life through narrative synthesis and random-effects meta-analysis. A sensitivity analysis assessed how sex differences varied by pain type, pain definition, and geographic region. **Results**: Eighteen eligible articles provided information on CP prevalence by sex and demonstrated variation according to pain type. All but three studies found a higher prevalence of CP in females than males. Based on a random-effects meta-analysis of eight studies, the overall relative risk (RR) was 1.16 (95% CI: 1.11–1.21) for females compared with males, with no evidence of heterogeneity. However, in subgroup analyses, the RR was lower for generic CP (RR = 1.16, 95% CI: 1.11–1.21) than for fibromyalgia (RR = 3.13, 95% CI: 1.22–8.04). **Conclusions**: Our review found that females are more likely to experience CP at mid-life, although the RR was small. Larger sex differences may be observed for fibromyalgia, but the small sample sizes highlight the need for larger studies to provide more precise estimates of different types of pain.

## 1. Introduction

Chronic pain (CP) is a global health challenge of increasing prevalence [1,2] and impact [3,4,5]. Prevalence estimates range between 7.1% [6] and 56%, with women overall being more likely to experience CP [7,8,9,10,11], but prevalence may differ according to life stage. Existing reviews have studied sex differences in prevalence during adolescence [12,13] and older age [11,14,15,16]. However, there is a lack of robust reviews examining the sex differences in CP in mid-life, a period variously defined between the ages of 40 and 65 [17,18,19,20,21], when socioeconomic and physical growth are often met with social stress connected to typically increased socioeconomic responsibilities [22,23,24,25,26] and physiological changes [1,18,27]. Mid-life is a period of life when CP can profoundly affect socioeconomic participation through a person’s ability to work [2,28] and to thrive [29,30]; hence, it should be a topic of interest in public health and political agendas. Moreover, mid-life has additional challenges for females, which could contribute to an exacerbation of the differences in CP distribution; there are gender-based expectations, which often assign caregiving responsibilities to women [31], along with physical changes like menopause [32,33]. Understanding sex differences in CP at mid-life from a population perspective could help direct sex-specific prevention and intervention programmes in this age group, with the aim of full socioeconomic participation.

Previous systematic reviews of sex differences in CP prevalence in adults have spanned a wide age range and have lacked differentiation between the different life stages [7,8,11,34,35,36,37]. Mansfield et al. (2016), for example, identified that prevalence of chronic widespread pain (CWP) was higher in women than men over 40, but this included older age [35], so the results are not relevant to a study of mid-life. Meanwhile, a review by Fayaz et al. (2016) [7] of adults reported higher prevalence of generic CP in females regardless of CP type but was also unable to comment on trends throughout the life course. One recent systematic review aimed at providing CP prevalence estimates in Europe provided a narrative summary of two studies that referenced both age and sex together, but the analysis was limited, as the focus extended beyond the topic of the review [38]. Overall, the reviewed literature tends to stratify by either sex but not age or by age but not sex, and so sex differences in CP, specifically during mid-life, have not been estimated. This systematic review aims to fill this gap in the knowledge by assessing the prevalence of CP in females and in males and the differences in CP prevalence between males and females in the general population aged 40–60 years. In addition, we investigated variations in sex differences by CP type (generic, regional, widespread, fibromyalgia), chronicity threshold, and geographic region.

## 2. Materials and Methods

A review protocol was designed following the Preferred Reporting Items for Systematic Reviews and Meta-Analyses Protocols (PRISMA-P) [39] guidelines and was registered with PROSPERO (ID: CRD42021295895) (available in the Appendix A). This was subsequently independently peer-reviewed and published [40].

### 2.1. Search Strategy

We searched MEDLINE (accessed through Web of Science as an interface), EMBASE, AMED, and PSYCHinfo (accessed through Ovid as an interface) for entries up to 10 January 2022. The search strategy was based on CP terms, study terms, moderators, and limits (full search strategy provided in the Appendix A) and was piloted to ensure the inclusion of key articles. The reference lists of eligible papers were screened to identify any additional publications.

### 2.2. Eligibility

We included studies published in peer-reviewed journals up to 10 January 2022 which, for those aged 40–60 years, provided CP prevalence in each sex, or the numbers from which the prevalence of CP for each sex could be derived, or an estimate of the sex differences in CP (e.g., relative risk (RR), odds ratio (OR), risk difference). Eligible studies used samples selected from the general population samples, had a clearly stated CP definition which matched the International Association for the Study of Pain (IASP) definition of pain lasting longer than three months [41], including both regional and/or widespread CP, clearly stated the country in which data was collected, and were written in English. Studies were excluded if they were reviews, conference proceedings, editorials or letters, or if they reported on samples of specific groups (e.g., clinical samples, population minorities).

### 2.3. Screening and Data Extraction

Two reviewers (CB and JP) screened all the search results by title and abstract and carried out the subsequent full-text screening. Any differences were resolved by discussion and consultation with a third member of the team (RH). Four reviewers (CB, JP, NR and RH) completed the data extraction using a purpose-designed data extraction form (Appendix A), with two reviewers working independently on each paper.

Information extracted included citation, the definition of chronic pain, country (and associated UN geographical region), and whether sex or gender was used. Data extracted included counts for CP and non-CP for each sex, sample size for males and females, prevalence of CP by sex and any estimates of sex difference (odds ratio, risk ratio or difference in prevalence) for the relevant age group. Where prevalence was only provided on figures, we used the WebPlotDigitizer tool (Autometris.io) to extract the information. Some studies provided information for more than one age group within the range of 40–60 years and data were extracted for all relevant groups. The geographical region of each study was allocated based on the United Nations (UN) region classification [42].

### 2.4. Quality-of-Evidence and Risk of Bias Assessment

Study quality was assessed by two reviewers (CB and JP), using a tool for risk of bias assessment for prevalence studies which considers internal and external validity and scores studies as a low, moderate, or high risk of bias [43]. The tool relies on a ten-item quality assessment, which includes the following questions:-Was the study’s target population a close representation of the national population in relation to relevant variables?-Was the sampling frame a true or close representation of the target population?-Was some form of random selection used to select the sample, OR was a census undertaken?-Was the likelihood of nonresponse bias minimal?-Were data collected directly from the subjects (as opposed to a proxy)?-Was an acceptable case definition used in the study?-Was the study instrument that measured the parameter of interest shown to have validity and reliability?-Was the same mode of data collection used for all subjects?-Was the length of the shortest prevalence period for the parameter of interest appropriate?-Were the numerator(s) and denominator(s) for the parameter of interest appropriate?

This tool has high interrater agreement, and it has previously been used in systematic reviews of pain prevalence [44]. Discrepancies were resolved by discussion and consultation with a third member of the team (RH).

### 2.5. Analysis

A narrative synthesis followed the Economic and Social Research Council Methods Programme guidelines [45], with a focus on exploring the prespecified sources of heterogeneity.

Estimates of CP prevalence by sex were calculated from counts and samples size where available. Meta-analysis was conducted using the Stata 17 (StataCorp. College Station, TX, USA). A random-effects meta-analysis was used to combine estimates of risk ratios for the sex difference in CP. The I^2^ value was calculated to assess the extent of heterogeneity in estimates, where values above 75% indicating high heterogeneity [46]. Pre-specified subgroup analyses were performed to investigate heterogeneity related to (i) geographic region (United Nations (UN) regional classification), the (ii) threshold for pain chronicity (in months), and (iii) pain type (e.g., generic, widespread, regional). A further subgroup analysis was carried out to assess whether there was heterogeneity according to the risk of bias in each study. The analyses were repeated with risk difference instead of relative risk. In addition, publication bias was assessed through the LFK index and Doi plot [47].

## 3. Results

### 3.1. Search Results

The study selection process is presented in Figure 1 and follows the Preferred Reporting Items for Systematic Reviews and Meta-Analyses (PRISMA) 2020 guidelines [48]. The search returned 5564 papers from Ovid and 3558 from Web of Science. After removing duplicates, the title and abstract of 7457 records were screened resulting in 254 full papers being retrieved to assess eligibility. Eighteen studies met the eligibility criteria. Of the eligible studies, eight presented data suitable for meta-analysis.

### 3.2. Study Characteristics

Eligible studies were published between 1994 and 2020 and the overall sample sizes varied from 600 to 27,035. All studies were cross-sectional and samples were recruited from the general population. One study was from Africa [49], four from Asia [50,51,52,53], seven from Europe [54,55,56,57,58,59,60], three from Northern America [61,62,63], and three from Oceania [53,60,63]. There were no eligible studies from Latin America and the Caribbean. The studies encompassed different types of CP: thirteen described generic CP—CP with no specifier [49,50,51,54,55,56,57,58,59,61,64,65,66]; one presented data for both chronic regional pain (CRP)—pain in one site only—and chronic widespread pain (CWP)—pain in multiple sites [52]; one presented data for CRP, CWP, and fibromyalgia [62]; and three presented data for fibromyalgia only [53,60,63]. Three studies used six months as the threshold for chronicity [61,65,66], while all others used three months. The categorization of sex was explicit (male, female) in eleven studies [49,50,52,53,54,56,60,62,63,64,66] while two studies used gender (woman, man) [58,59] and five studies used sex and gender interchangeably [51,55,57,61,65]. Studies that reported gender rather than sex did not provide a definition of gender. The full details of the eligible studies are summarized in Table 1 and Table 2. The Appendix A, with further details about sampling frames and chronic pain case definitions.

Pain prevalence for each sex was reported or could be calculated in all studies apart from one, and from these, the RR and risk difference were calculated. The remaining study only reported the difference in prevalence (risk difference) between females and males [59]. Of the eighteen eligible studies, eight provided information that was suitable for meta-analysis of RRs and risk differences [53,55,56,57,58,61,63,64]. The other ten studies did not provide the required information to calculate the standard error of the RR, which is required to weight estimates in the meta-analysis. Of the eight included in meta-analysis, three reported the counts for each group (i.e., females and males with CP, females and males with no CP) [53,61,63] which could be used directly in meta-analysis. For four other studies, the relevant counts could be derived from either prevalence and CP counts [58,64], prevalence and sample size [57], or prevalence and 95% confidence interval [64]. The remaining eleven studies reported percentages of participants with CP without providing the relevant sample sizes and/or counts with and without CP [49,50,51,52,54,60,62,65,66] or reported difference in prevalence but did not include counts for participants without CP [59]. Table 3 shows relative risk for each study.

### 3.3. Quality of Evidence and Risk of Bias

The risk of bias for each study is reported in Table 1. Eight studies were deemed at low risk of bias [50,52,56,58,61,63,64,66], five studies at moderate [49,51,54,57,65] and five studies at high risk [53,55,59,60,62]. Of the studies at high risk-of-bias three were from Europe, one from Asia and one from Northern America. The quality-of-evidence and risk of bias assessment are available in the Appendix A.

### 3.4. Narrative Analysis

CP prevalence in females ranged between 3% [62] and 76% [54] and for males between 0% [60] and 69% [54] (Table 2, Figure 2). The highest prevalence was in the generic CP type reported in a sample from rural Sweden [54], while the two lowest prevalence estimates were in the four studies of fibromyalgia [53,60,62,63].

Overall, the generic CP type in females ranged between 14% [51] and 76% [54], and in males between 9% [51] and 69% [54]—the lower values in both sexes were derived from an Asian study [51] while the higher values were from a European study [54]. CRP prevalence ranged between 14% [52] and 25% [62] in females and between 16% [52] and 23% [62] in males, where the lower values in both sexes were again derived from an Asian study [52] while the higher values were from a European study [62]. CWP prevalence ranged between 15% [52] and 21% [62] in females, and between 4% [52] and 13% [62] in males. Finally, fibromyalgia ranged between 3% [62] and 20% [53] in females, and 0% [60] and 9% [53] in males—the lower values were from a northern American and a European study [60,62], and the higher values were from an Asian study [53].

CP was more prevalent in females than in males in sixteen of the eighteen eligible studies [49,50,51,52,53,54,55,56,57,58,59,60,63,64,65,66] with the RR ranging from 1.04 for generic CP [54] to 14.0 for fibromyalgia [60]. There were two exceptions in which CP was more common in men. First, Andersson et al. (1994) reported a higher prevalence of generic CP in males at ages 55–59 (RR = 0.91), although not at ages 45–54 [54]. Second, in Buskila et al. (2000) CRP was more prevalent in males at ages 41–50 (RR = 0.88) and 51–60 (RR = 0.95) [52].

CWP, defined variously as CP in more than one site [52] or as pain above and below the waist, on both the right and left side of the body, and at an axial location consistent with the ACR 1990 definition [67] was considerably more common in females in two studies [52,62]. There was a four-fold greater prevalence in the study with the greatest difference (RR = 3.75 in 41–50-year-olds and RR = 3.50 in 51–60-year-olds) [52]. In the second study, the RR was more conservative (RR = 1.69 in 40–59-year-olds) but higher than that for the CRP group in the same study (RR = 1.08 in 40–59-year-olds) [62]. Similarly, fibromyalgia was consistently more prevalent in females based on four studies [53,60,62,63], with the RR varying between 14.00 [60] and 2.18 [53] (Table 3, Figure 3). Please note that although the same studies are used, we do not double-counted participants.

### 3.5. Meta-Analysis

The meta-analysis included 17,304 participants from eight studies [53,55,56,57,58,61,63,64]. Of the eight studies, six measured generic CP [55,56,57,58,61,64] and two measured fibromyalgia [53,63] (Table 4).

Overall, the RR of CP in mid-life for women compared to men was estimated to be 1.16 (95% confidence interval (CI) 1.11–1.21) (Table 5). There was no evidence of heterogeneity in estimates across studies with an I^2^ value of 0. A forest plot representing RR is available in the Appendix A.

#### 3.5.1. Subgroups Analyses

Subgroup analysis by geographic region demonstrated overall RRs that was similar in the four studies from Europe (1.18, 95% CI 1.09–1.27) [55,56,57,58] and a single study from Oceania [64], but the RR values from a single study in Asia [53] and from studies in Northern America [63] were higher. However, the study from Asia and one of those from Northern America were on fibromyalgia [53,63]. (Table 5). Therefore, it is likely pain type rather than country that underlies these differences. Due to the small number of countries represented in the meta-analysis we did not carry out subgroup analysis by human development index metric–an index of population wealth. Forest plots representing the subgroup analysis are available in Appendix A.

In the subgroup analysis by pain threshold there was only one study that used the six months chronicity threshold [61]. The overall RR for the three-month threshold was 1.17 (95% CI 1.14–1.24), while the six-month threshold study which gave an RR of 1.16 (95% CI 1.08–1.24).

The subgroup analysis by pain type showed a greater RR for fibromyalgia (3.13, 95% CI 1.22–8.04) [53,63] compared to generic CP (1.19, 95% CI 1.13–1.25) [55,56,57,58,61,64]. Heterogeneity was low in both groups, although the two fibromyalgia studies were small and produced estimates with wide confidence intervals.

#### 3.5.2. Heterogeneity by Risk of Bias

In the eighteen studies, those with a low risk of bias [50,52,56,58,61,63,64,66] had an RR ranging between 0.88 and 6.27, the studies with a moderate risk had an RR [49,51,54,57,65] ranging between 0.91 and 1.42, and those with a high risk [53,55,59,60,62] had an RR ranging between 1.08 and 14.00.

In studies included in the meta-analysis, the overall RR was higher in two studies with a high risk of bias [53,55] (1.43, 95% CI 0.79–2.59), compared to the five studies with a low risk of bias (1.16, 95% CI 1.10–1.22) [56,58,61,63,64] and the single study with a moderate risk of bias (1.14, 95% CI 1.18–1.61) [57]. In the high-risk-of-bias group, one study focused on fibromyalgia with a high RR estimate [53]. The variation in RR was therefore again likely driven by differing pain types (Table 5).

The small number of studies also meant that we were unable to meaningfully assess publication bias using the funnel plot, but the LFK index and a DOI plot (Appendix A) suggested evidence of publication bias. The LFK index has been suggested as an improved approach to assessing publication bias, but a recent simulation study has suggested that its performance does depend on the number and size of studies and the between-study heterogeneity [68].

#### 3.5.3. Supplementary Analysis: Overall Risk Difference of Chronic Pain Between Males and Females

The meta-analysis of risk difference (RD) in CP prevalence between sexes was conducted on the same eight studies included in the primary meta-analysis [53,55,56,57,58,61,63,64]. The RD between the sexes varies from 0.03 (95% CI: 0.01–0.05) [64] to 0.10 (95% CI: 0.01–0.20) [53] and the pooled difference is 0.05 (95% CI: 0.03–0.07). This analysis shows some heterogeneity (I^2^ 31.15%) in contrast to the analysis of RR. RD was highest in Asia (0.10, CI 95% 0.03–0.07) in a single fibromyalgia study [53] and lowest in Oceania (0.03, CI 95% 0.01–0.05) in a generic CP study [64]. RD is higher in the fibromyalgia sub-type studies (0.06, CI 95% 0.03–0.07) [53,63] compared with the CP. There was no significant difference according to chronicity threshold. The subgroup analysis by risk of bias showed slightly higher estimates in high-risk studies, similar to the main RR analysis. A forest plot describing the risk difference analysis and subgroups analyses are available in the Appendix A.

## 4. Discussion

The overall estimate from our meta-analysis indicates that females are 16% (RR = 1.16, 95% CI 1.11–1.21) more likely to have CP than males, with the RR being considerably higher in studies of fibromyalgia. However, the limited number of studies, and small number of fibromyalgia events within studies means that investigation of heterogeneity was limited. While the meta-analysis was based on only eight out of eighteen studies, in our narrative review, we found a wide range of prevalence of CP in both women and men at mid-life, reflecting differences both in pain type and geographic location, but heterogeneity for the sex difference was low throughout.

Our review builds on the previous literature of differences in CP prevalence between sexes. The consistently higher RR of CP in females in mid-life is in line with findings from puberty onwards, suggesting underlying sex and/or gender mechanisms that are ongoing or may be replicated at different life and reproductive stages. A review of generic CP in adolescents and young adults shows that prevalence is 8.6% in females and 6.2% in males [69], with an overall RR of 1.39. In a review of over-75-year-olds, prevalence was 61.5% (95% CI 43.5–64.7) in females and 54.1 (95% CI 43.5–64.7) in males [56] with an overall RR of 1.14. In a review on over-65-year-olds, CP prevalence was 42.5% (95% CI 39.0–46.0) in females and 33.3% (95% CI 29.6–37.1) in males [61] with an overall RR of 1.28. There are no significant differences between the RR of the different reviews; however, CP prevalence ranges are notoriously wide, so variations in RR could be an effect of methodological variation in the different studies. Thus, comparisons should be interpreted accordingly.

Indeed, the variation in prevalence within the studies reviewed in this paper is consistently high, like other reviews of CP prevalence [7,34,35,69,70]. This variation in prevalence might be due to the way in which study participants engage with questions about pain chronicity, which can also manifest in geographic differences. Moreover, some studies asked about chronic or recurring pain—with recurring pain being poorly defined and opening the risk of inclusion of non-CP cases. Lastly, some acute injuries produce pain that may last longer than twelve weeks, which means these may also be included as generic CP cases. Considering that pain is a normal part of daily life, it is important for future studies to focus on combined measurements of chronicity and impact.

One significant finding is that CP prevalence varied according to pain type, with the CWP and fibromyalgia types showing much higher RRs than for CRP [53,60,62,63]. In particular, robust studies of primary data have described a higher prevalence of fibromyalgia in females [71], although, to the authors’ knowledge, there are no systematic reviews of fibromyalgia prevalence by sex and age. In line with the possibility that pain type is a driver of heterogeneity in the sex difference, other reviews have also pointed out that differences in prevalence between the sexes varied by pain type. In children and adolescents, migraine, abdominal pain and musculoskeletal pain were more common in females, while back pain was not [13]. Another review of CP prevalence in young adults found persisting high heterogeneity in subgroup analysis by pain type, arguing this may be driven by methodological differences between studies [69], but the study did not investigate sex difference within pain types. Our results are supported by a further systematic review which shows both prevalence and RD of CWP peaking in mid-life [35]—in contrast to other pain types [7]. Our meta-analysis, however, did not include studies of CWP, and only two studies of fibromyalgia. Fibromyalgia is a form of CWP that requires a stringent clinical assessment to rule out other diagnoses [67] which makes it harder to assess in epidemiological studies. Although the two fibromyalgia studies have large RRs for the sex difference, the prevalence is low in both sexes and, further, due to their relatively small sample sizes, the estimated RRs have wide confidence intervals [53,63]. Studies with small sample sizes can lead to bias in RR calculations because within-study variances are treated as true variances, leading to sampling error [72] and greater estimates known as small-study effects [73]. Random-effect meta-analysis accounts for between-study heterogeneity and differences in sample sizes by assigning smaller weight to studies with small sample sizes, meaning that such studies contribute little information to the meta-analysis [74]. As such, our overall RR is weighted heavily towards the larger studies of generic CP. Larger studies of fibromyalgia as well as CWP are required to provide more precise estimates.

Our inclusion criteria meant that all studies included in the review had a homogeneous definition of CP lasting over three months. Included studies defined CP according to the 2015 IASP definition, even if they were published prior to this. However, there are methodological fallacies with estimating CP by chronicity threshold as this is not representative of the complex physical, functional, and emotional presentation of people living with CP. While it is possible for many to have intermittent, low-grade pain over prolonged periods of time this is not equivalent to a clinical CP presentation or high-impact CP. Hence, the generic CP phenotype might be less relevant in describing impactful CP, whereas CWP and fibromyalgia can capture to some degree the extent of physical involvement. Although we found little heterogeneity according to our pre-specified subgroups, it was not possible to comment on the role of geographic location, or threshold for chronicity because in the included studies they are confounded with pain type (i.e., all studies in Europe are of CP and the single study from Asia is fibromyalgia).

Our systematic review highlighted methodological limitations of existing studies in CP. Future studies should address the measurement of sex differences in all pain phenotypes in mid-life, something which may improve in the future with the uptake in research of a unified CP classification system [75]. The addition of categories such as ‘high impact’ CP may also be beneficial in characterizing the sample; this was not available in the studies reviewed in this paper. Age bands were often difficult to harmonize within the review process [7,34,35], so ensuring that banding is structured around life stages will facilitate life-course research. There was a failure to characterize the sample and to report the numbers with and without chronic pain within each sex (or the number with CP and numbers of males and females) to allow reviewers to calculate RR and RD and their confidence intervals. In line with guidelines for reporting of observational studies, studies of CP should include the sample sizes for main and any subgroup analyses. Lastly, sex and gender were often used interchangeably. More clarity should be given to the measures used to collect this information and to explain the rationale for choosing sex or gender classification to contribute to understanding how each contribute to pain prevalence.

### Strengths and Limitations

This review followed a robust and replicable process based on best practice for systematic reviews. The search was conducted across databases relevant to the research topic, and the multiple reviewers ensured the risk of error in study inclusion/exclusion and data extraction was minimized.

The limited number of studies included in our meta-analysis reflect the small number of studies providing relevant information for both age and sex. While the RR could be derived for most studies, few studies provided information required to allow the calculation of the standard error. We highlight the need for the consistent reporting of sex prevalence data in the CP literature. The number of studies was also limited by the selection criteria for definition of CP to standards of the IASP—which we believe is a key strength of our review. Others have commented on the heterogeneity in CP prevalence within the literature and attributed it to heterogeneity in CP definition [76].

The exclusion of studies in languages other than English will have limited the range of studies included and may have particularly limited our planned subgroup analysis by geographic region; future reviews may want to consider additional databases like LILACS, as well as grey literature. Furthermore, our original review protocol set out to do further geographic analysis by the WHO region classification and the Human Development Index (HDI) for each country, but the small number of studies meant this would not add value.

Finally, the small number of studies suitable for meta-analysis means that the overall estimates of sex difference and subgroup analyses should be interpreted with caution. Given the limited number of studies, we were unable to conduct meta-regressions within the sensitivity analysis [77] and this is also relevant to the interpretation of the risk of bias assessment.

## 5. Conclusions

The risk of CP was slightly, and consistently, higher in women than men, with a suggestion of a much greater sex difference in the limited number of studies investigating fibromyalgia. Further larger studies are required to estimate how sex differences vary by type. Understanding these differences can lead to targeted research around the mechanisms driving inequality and sex-aware interventions with important socioeconomic consequences.

## Figures and Tables

**Figure 1 biomedicines-13-02523-f001:**
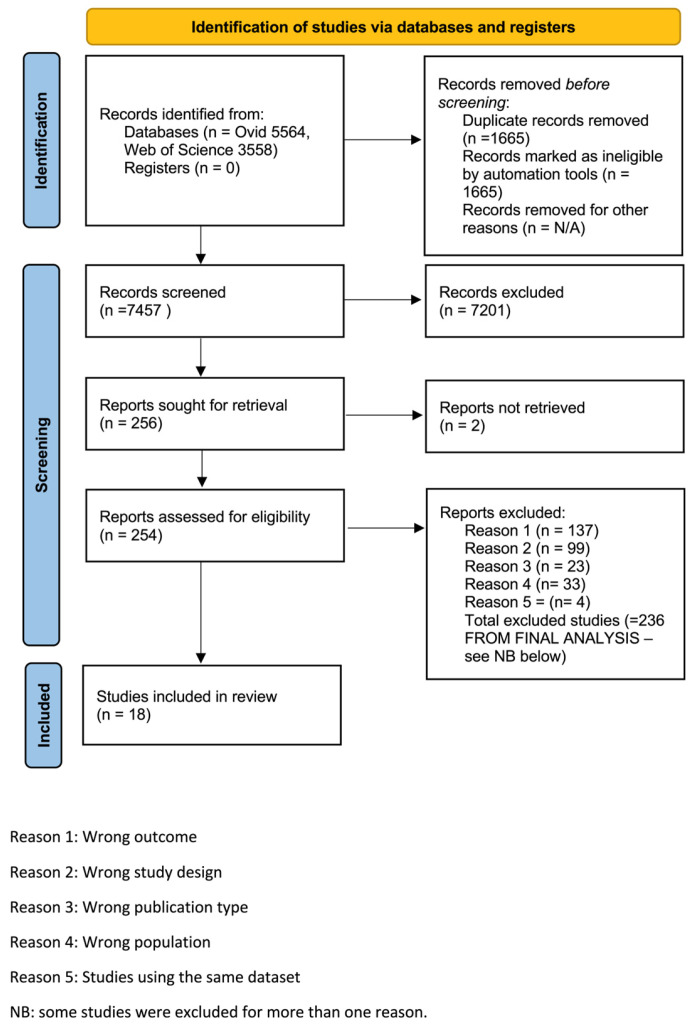
Study identification flow diagram.

**Figure 2 biomedicines-13-02523-f002:**
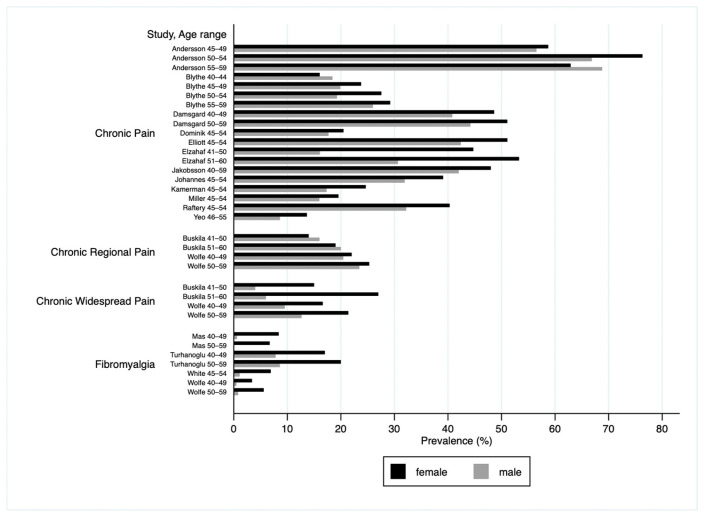
Prevalence by pain type.

**Figure 3 biomedicines-13-02523-f003:**
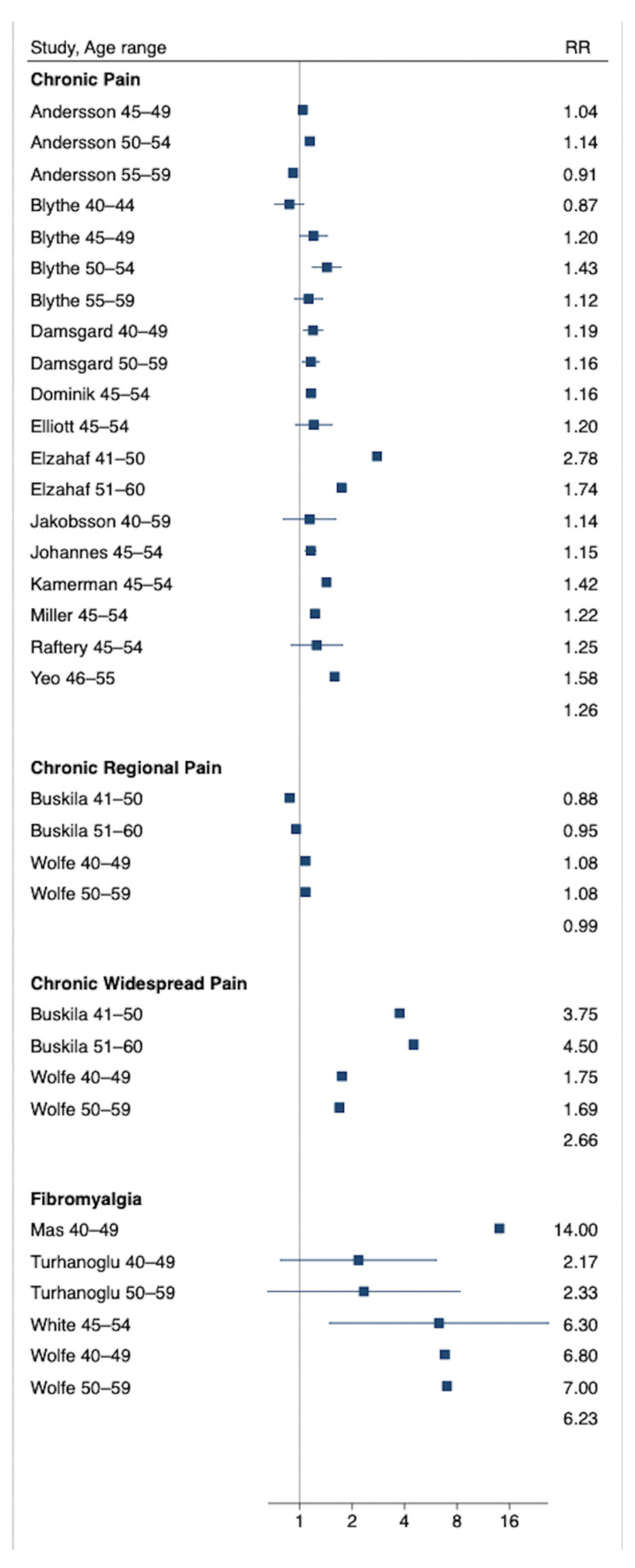
Relative risk (and 95% confidence intervals in studies where there is enough information to calculate them) by pain type. Note: Where no 95% confidence interval is plotted, the study did not provide adequate information for them to be calculated, i.e., did not provide information from which the standard error could be calculated. Therefore, no overall estimates are provided.

**Table 1 biomedicines-13-02523-t001:** Summary of studies included and their sample age, geographic region, pain definition, chronicity threshold, sex/gender measurement and risk of bias.

Study	Analytic Sample Age	Country, UN Geographic Region	Pain Definition	Chronicity Threshold	Sex or Gender	Risk of Bias
Andersson et al. (1994) [54]	45–59	Southern Sweden—Europe	Chronic pain A	Over 3 months	sex	Moderate
Blyth et al. (2001) [64]	40–59	New South Wales, Australia—Oceania	Chronic pain	Over 3 months	sex	Low
Buskila et al. (2000) [52]	41–50	Yeruham, Israel—Asia	Chronic regional pain A	Over 3 months	sex	Low
			Chronic widespread pain B	Over 3 months		
Damsgård et al. (2020) [55]	40–59	North Norway—Europe	Chronic pain	Over 3 months	interchangeable	High
Dominick et al. (2011) [65]	45–54	New Zealand—Oceania	Chronic pain	Over 6 months	Interchangeable	Moderate
Elliott et al. (2002) [56]	45–54	Grampian, Scotland—Europe	Chronic pain	Over 3 months	Sex	Low
Elzahaf et al. (2016) [50]	41–60	Tripoli, Benghazi and Sabha, Libya—Asia	Chronic pain	Over 3 months	Sex	Low
Jakobsson (2010) [57]	45–59 (160, M 46.3%, F 53.8%)	Province of Skåne, Sweden—Europe	Chronic pain	Over 3 months	Interchangeable	Moderate
Johannes et al. (2010) [61]	45–54	United States of America—Northern America—0.926	Chronic pain A	Over 6 months	Interchangeable	Low
Kamerman et al. (2020) [49]	45–54	South Africa—Africa	Chronic pain A	Over 3 months	Sex	Moderate
Mas et al. (2008) [60]	40–59	Spain—Europe—0.904	Fibromyalgia C	Over 3 months	Sex	High
Miller et al. (2017) [66]	45–54	Australia—Oceania—0.944	Chronic pain	Over 6 months	Sex	Low
Raftery et al. (2011) [58]	45–54	Ireland—Europe	Chronic pain	Over 3 months	Gender	Low
Rustøen et al. (2005) [59]	40–59	Norway—Europe	Chronic pain	Over 3 months	Gender	High
Turhanoğlu et al. (2008) [53]	40–59	Turkey—Asia	Fibromyalgia C D	Over 3 months	Sex	High
White et al. (1999) [63]	45–54	Canada—Northern America—0.929	Fibromyalgia C D	Over 3 months	Sex	Low
Wolfe et al. (1995) [62]	40–59	Wichita, United States of America—Northern America—0.926	Chronic regional pain E	Over 3 months	Sex	High
			Chronic widespread pain F			
			Fibromyalgia C D			
Yeo and Tai (2009) [51]	46–55	Singapore—Asia	Chronic pain A	Over 3 months	Interchangeable	Moderate

(A) Chronic pain: both persistent and recurring pain lasting for the given duration. (B) Chronic widespread pain: pain present in more than one site for the given duration. (C) Fibromyalgia: defined by the American College of Rheumatology (ACR) classification criteria: (1) widespread pain on the day of the interview, defined as (a) pain in at least one point in both the right and the left half of the body, above and below the waist, and axial pain; (b) which is greater than 1 on a visual analogue scale (0–10); (c) which has been present for more than three months; and (d) which is unrelated to cancer or traumatism; and (2) finding of 11 out of 18 possible tender points on examination as established by the ACR. (D) Fibromyalgia was assessed by a clinician based on the ACR classification criteria. (E) Chronic regional pain: defined as pain for at least 1 week in past months in upper and lower quadrants lasting for longer than 3 months. (F) Chronic widespread pain: defined as pain experienced for 3 or more months above and below the waist, on both the right and left side of the body, and at an axial location (spine, chest, sternum, upper or lower back), in line with the ACR definition of chronic widespread pain.

**Table 2 biomedicines-13-02523-t002:** Summary of studies included their pain prevalence data and counts where available or derived where possible.

Study	Sample Size in Relevant Age Range (Total *n* of Original Study)	Pain Type	Pain Prevalence (%, 95% CI)		Number Reporting Pain		Sample Size	
			Female	Male	Female	Male	Female	Male
Andersson et al. [54] (1994)	639 ^A^ (1609)	Chronic pain	Age 45–49: 58.71 ^B^ Age 50–54: 76.31 ^B^ Age 55–59: 62.90 ^B^	Age 45–49: 56.51 ^B^ Age 50–54: 66.85 ^B^ Age 55–59: 68.79 ^B^	Not provided	Not provided	Not provided	Not provided
Blyth et al. (2001) [64]	6506 (17,543)	Chronic pain	Age 40–44: 16.05 ^B^ Age 45–49: 23.78 ^B^ Age 50–54: 27.56 ^B^ Age 55–59: 29.22 ^B^	Age 40–44: 18.44 ^B^ age 45–49: 19.87 ^B^ Age 50–54: 19.28 ^B^ Age 55–59: 25.98 ^B^	Age 40–44: 187 Age 45–49: 210 Age 50–54: 211 Age 55–59: 217	Age 40–44: 161 Age 45–49: 165 Age 50–54: 137 Age 55–59: 139	Age 40–44: 1165 ^C^ Age 45–49: 883 ^C^ Age 50–54: 766 ^C^ Age 55–59: 743 ^C^	Age 40–44: 873 ^C^ Age 45–49: 830 ^C^ Age 50–54: 711 ^C^ Age 55–59: 535 ^C^
Buskila et al. (2000) [52]	Not provided (2210)	Chronic regional pain	Age 41–50: 14.00 Age 51–60: 19.00	Age 41–50: 16.00 Age 51–60: 20.00	Not provided	Not provided	Not provided	Not provided
		Chronic widespread pain	Age 41–50: 15.00 Age 51–60: 27.00	Age 41–50: 4.00 Age 51–60: 6.00	Not provided	Not provided	Not provided	Not provided
Damsgård et al. (2020) [55]	2737 (5546)	Chronic pain	Age 40–49: 48.62 Age 50–59: 51.07	Age 40–49: 40.81 Age 50–59: 44.19	Age 40–49: 353 ^D^ Age 50–59: 429 ^D^	Age 40–49: 211 ^D^ Age 50–59: 289 ^D^	Age 40–49: 726 ^D^ Age 50–59: 840 ^D^	Age 40–49: 517 ^D^ Age 50–59: 654 ^D^
Dominick et al. (2011) [65]	Not provided (12,488)	Chronic pain	Age 45–54: 20.50 (17.6–23.5) ^E^	Age 45–54: 17.70, (14.5–21.0) ^E^	Not provided	Not provided	Not provided	Not provided
Elliott et al. (2002) [56]	326 (1608)	Chronic pain	Age 45–54: 51.1 (43.8–58.4)	Age 45–54: 42.4 (34.3–50.4)	Age 45–54: 92 ^F^	Age 45–54: 62 ^F^	Age 45–54: 180 ^F^	Age 45–54: 146 ^F^
Elzahaf et al. (2016) [50]	332 (1212)	Chronic pain	Age 41–50: 44.72 ^B^ Age 51–60: 53.27 ^B^	Age 41–50: 16.08 ^B^ Age 51–60: 30.65 ^B^	Not provided	Not provided	Not provided	Not provided
Jakobsson (2010) [57]	160	Chronic pain	Age 45–59: 48	Age 45–59: 42	41 ^G^	31 ^G^	86	74
Johannes et al. (2010) [61]	6687 (27,035)	Chronic pain	Age 45–54: 39.1 (36.3–42.0) ^E^	Age 45–54: 31.9 (28.7–35.2) ^E^	1583	815	4194	2493
Kamerman et al. (2020) [49]	1373 (10,336)	Chronic pain	Age 45–54: 24.64 (21.34–28.57) ^B,E^	Age 45–54: 17.32 (13.39–22.05) ^B,E^	Not provided	Not provided	Not provided	Not provided
Mas et al. (2008) [60]	Not provided (2192)	Fibromyalgia	Age 40–49: 8.4 Age 50–59: 6.7	Age 40–49: 0.6 Age 50–59: 0	Not provided	Not provided	Not provided	Not provided
Miller et al. (2017) [66]	Not provided (16,412)	Chronic pain	Age 45–54: 19.54 ^B^	Age 45–54: 15.99 ^B^	Not provided	Not provided	Not provided	Not provided
Raftery et al. (2011) [58]	242 (1204)	Chronic pain	Age 45–54: 40.3	Age 45–54: 32.2	Age 45–54:50	Age 45–54: 38	Age 45–54: 124 ^C^	Age 45–54: 118 ^C^
Turhanoğlu et al. (2008) [53]	187 (600)	Fibromyalgia	Age 40–49: 17.0 Age 50–59: 20.0	Age 40–49: 7.8 Age 50–59: 8.6	Age 40–49: 9 Age 50–59: 7	Age 40–49: 5 Age 50–59: 3	Age 40–49: 53 Age 50–59: 35	Age 40–49: 64 Age 50–59: 35
White et al. (1999) [63]	459	Fibromyalgia	Age 45–54: 6.9 (6.2–7.6)	Age 45–54: 1.1 (0.4–2.9)	Age 45–54: 19	Age 45–54: 2	Age 45–54: 276	Age 45–54: 183
Wolfe et al. (1995) [62] ^E^	912 (3006)	Chronic regional pain	Age 40–49: 22.01 (19.4–24.6) ^E^ Age 50–59: 25.29 (22.3–28.3) ^E^	Age 40–49: 20.44 (17.8–22.8) ^E^ Age 50–59: 23.45 (20.4–26.4) ^E^	Not provided	Not provided	Not provided	Not provided
		Chronic widespread pain	Age 40–49: 16.61 (14.0–19.0) ^E^ Age 50–59: 21.39 (18.2–24.6) ^E^	Age 40–49: 9.51 (7.6–11.4) ^E^ Age 50–59: 12.65 (10.1–15.1) ^E^	Not provided	Not provided	Not provided	Not provided
		Fibromyalgia	Age 40–49: 3.4 (1.4–4.6) ^E^ Age 50–59: 5.6 (3.2–8.0) ^E^	Age 40–49: 0.5 (0.0–1.0) ^E^ Age 50–59: 0.8 (0.0–1.7) ^E^	Not provided	Not provided	Not provided	Not provided
Yeo and Tai (2009) [51]	Not provided (4141)	Chronic pain	Age 46–55: 13.65 ^B^	Age 46–55: 8.62 ^B^	Not provided	Not provided	Not provided	Not provided

^A^ Calculated from total *n* and % in each age group. ^B^ Extracted by researchers from figure. ^C^ Sample size derived from published prevalence and pain count data. ^D^ Calculated by adding numbers in two different ethnic groups. ^E^ Weighting applied to calculate prevalence and 95% confidence interval. ^F^ Sample size calculated from 95% confidence interval of the prevalence and number with pain calculated from prevalence and derived sample. ^G^ Number with pain derived from published prevalence and sample size.

**Table 3 biomedicines-13-02523-t003:** Summary of studies included and their relative risk according to pain type in 18 studies (with confidence intervals where counts are available to allow calculation).

	Relative Risk (95% Confidence Interval Where Counts Are Available to Allow Calculation) for Female vs. Male Pain			
Study	Generic Chronic Pain	Chronic Regional Pain	Chronic Widespread Pain	Fibromyalgia
Andersson et al. (1994) [54] A	Age 45–49: 1.04 Age 50–54: 1.26 Age 55–59: 0.88			
Blyth et al. (2001) [64] B	Age 40–44: 0.87 (0.72, 1.05) Age 45–49: 1.20 (1.00, 1.43) Age 50–54: 1.43 (1.18, 1.73) Age 55–59: 1.12 (0.95, 1.35)			
Buskila et al. (2000) [52] A		Age 41–50: 0.88 Age 51–60: 0.95	Age 41–50: 3.75 Age 51–60: 4.5	
Damsgård et al. (2020) [55] B	Age 40–49: 1.19 (1.05, 1.35) Age 50–59: 1.16 (1.04, 1.29)			
Dominick et al. (2011) [65] A C	Age 45–54: 1.17			
Elliott et al. (2002) [56] B	Age 45–54: 1.20 (0.95, 1.53)			
Elzahaf et al. (2016) [50] A	Age 41–50: 2.81 Age 51–60: 1.71			
Jakobsson (2010) [57] B	Age 40–59: 1.14 (0.80, 1.61)			
Johannes (2010) [61] B C	Age 45–54: 1.16 (1.08, 1.24)			
Kamerman (2020) [49] A C	Age 45–54: 1.47			
Mas (2008) [60] A				Age 40–49: 8.0
				Age 50–59: N/A D
Miller (2017) [66] A	Age 45–54: 1.25			
Raftery (2011) [58] B	Age 45–54: 1.25 (0.89–1.76)			
Turhanoğlu (2008) [53] B				Age 40–49: 2.17 (0.78, 6.09) Age 50–59: 2.33 (0.66, 8.30)
White (1999) [63] B				Age 45–54: 6.23 (1.49, 26.72)
Wolfe (1995) [62] A C		Age 40–49: 1.10 Age 50–59: 1.09	Age 40–49: 1.70 Age 50–59: 1.9169	Age 40–49: 3.0 Age 50–59: 6.0
Yeo (2009) [51] A	Age 46–55: 1.56			

(A) The relative risk (RR) is calculated from prevalence figures, with decimals rounded. (B) The RR was calculated from counts presented in the paper. (C) The prevalence data in the paper was weighted. (D) The RR cannot be calculated due to absent cases in men.

**Table 4 biomedicines-13-02523-t004:** Summary of studies included in meta-analysis, with prevalence and cases representing the total sample of the eligible age groups(s).

Study	Sample Size in Relevant Age Range (Total *n* of Original Study)	Pain Type	Pain Prevalence (%, 95% CI)		Number Reporting Pain		Sample Size	
			Female	Male	Female	Male	Female	Male
Blyth et al. (2001) [64]	6506 (17,543)	Chronic pain	Age 40–59: 23.19 A	Age 40–59: 20.41 A	Age 40–59: 825 A	Age 40–59: 602 A	Age 40–59: 3557 A	Age 40–59: 2949 A
Damsgård et al. (2020) [55]	2737 (5546)	Chronic pain	Age 40–59: 49.94 B	Age 40–59: 42.70 B	Age 40–59: 782 B	Age 40–59: 500 B	Age 40–59: 1566 B	Age 40–59: 1171 B
Elliott et al. (2002) [56]	326 (1608)	Chronic pain	Age 45–54: 51.1 (43.8–58.4)	Age 45–54: 42.4 (34.3–50.4)	Age 45–54: 92	Age 45–54: 62	Age 45–54: 180	Age 45–54: 146
Jakobsson (2010) [57]	160	Chronic pain	Age 45–59: 48	Age 45–59: 42	Age 45–59: 41	Age 45–59: 31	Age 45–59: 86	Age 45–59:
Johannes et al. (2010) [61]	6687 (27,035)	Chronic pain	Age 45–54: 39.1 (36.3–42.0)	Age 45–54: 31.9 (28.7–35.2)	Age 45–54: 1583	Age 45–54: 815	Age 45–54: 4194	Age 45–54: 2493
Raftery et al. (2011) [58]	242 (1204)	Chronic pain	Age 45–54: 40.3	Age 45–54: 32.2	Age 45–54: 50	Age 45–54: 38	Age 45–54: 124	Age 45–54: 118
Turhanoğlu et al. (2008) [53]	187 (600)	Fibromyalgia	Age 40–59: 18.18	Age 40–59: 8.08	Age 40–59: 16 B	Age 40–59: 8 B	Age 40–59: 88 B	Age 40–59: 99 B
White et al. (1999) [63]	459	Fibromyalgia	Age 45–54: 6.9 (6.2–7.6)	Age 45–54: 1.1 (0.4–2.9)	Age 45–54: 19	Age 45–54: 2	Age 45–54: 276	Age 45–54: 183

(A) Summary of age groups: 40–44, 45–49, 50–54, 55–59. (B) Summary of age groups: 40–49, 50–59.

**Table 5 biomedicines-13-02523-t005:** Results from the overall meta-analysis of CP prevalence and subgroup analyses.

	Studies	Sample Size	Pooled Estimate (RRR)	95% CI	I^2^
Primary analysis					
	8	17,304	1.16	1.11–1.21	0.00
Subgroup analysis by UN geographic region					
Africa	0	-	-	-	-
Asia	1	187	2.25	1.01–5.00	-
Europe	4	3465	1.18	1.09–1.27	0.00
Latin America and Caribbean	0	-	-	-	-
Northern America	2	7146	2.30	0.45–11.77	81.08
Oceania	1	6506	1.14	1.04–1.25	-
Subgroup analysis by CP type					
CP	6	16,658	1.16	1.04–1.25	0.02
Fibromyalgia	2	646	3.13	1.22–8.04	33.04
Subgroup analysis by threshold for chronicity					
3 months	7	10,617	1.17	1.10–1.24	0.00
6 months	1	6687	1.16	1.08–1.24	-
Subgroup analysis by risk of bias					
Low	5	14,220	1.16	1.10–1.22	0.00
Moderate	1	160	1.14	0.80–1.61	-
High	2	2924	1.43	0.79–2.59	60.81

## Data Availability

The search strategy allows for replicability in data collection. The corresponding author can provide the statistical code used for the analysis on request.

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
