# Peer review of "Sex Differences in the Prevalence of Chronic Pain in Mid-Life: A Systematic Review and Meta-Analysis"

_biomedicines, 2025, doi:10.3390/biomedicines13102523_

Round 1

Reviewer 1 Report

Comments and Suggestions for Authors

I have studied the manuscript entitled "Sex differences in the prevalence of chronic pain in mid-life: a systematic review and meta-analysis" by Borra C. et al.

The present manuscript deals with a health concern of major impact, which might be of interest not only for the specialists, but also for a broader readership. The research question retains some originality and worths a thourough meta-analysis.

The language is acceptable, though the manuscript would be benefited from a professional editing.

The text is generally well prepared. The methodology followed is suitable, though some improvements would be desirable.

Before considering publication, the authors are welcome to assess/discuss the following issues.

Major issues

1) Line 265: The authors are kindly suggested to use altrnative tools to estimate publication bias, such as Doi plot and LFK index (see: Shamim MA et al. Beyond the funnel plot: The advantages of Doi plots and prediction intervals in meta-analyses. Asian J Psychiatr. 2023, 84, 103550).

2) The overall prevalence difference observed could at least be partly attributed to the fact that fibromyalgia is much more prevalent among women (see: Cooksey R. et al. Exploring gender differences, medical history, and treatments used in patients with fibromyalgia in the UK using primary-care data: a retrospective, population-based, cohort study. The Lancet Rheumatology, 2022, 4, S20). The authors are kindly suggested to further discuss this key issue, by additionally implementing any proper data analysis.

Minor issues

1) Line 25 (Abstract): The authors are kindly suggested to define the abbreviation "FM".

2) Line 61: The authors are kindly suggested to provide the detailed search algorithm for at least one database.

3) Line 191 (Figure 3): The authors are kindly suggested to add 95% CI for each study as well as  subtotals for each subgroup.

Author Response

Dear Reviewer 1,

Thank you for your thoughtful input. Please find the response to the major issues:

1) Line 265: The authors are kindly suggested to use alternative tools to estimate publication bias, such as Doi plot and LFK index (see: Shamim MA et al. Beyond the funnel plot: The advantages of Doi plots and prediction intervals in meta-analyses. Asian J Psychiatr. 2023, 84, 103550).

Following your advice, we have added a Doi plot and LFK index. We have reflected this in the methods section (line 120-121, page 4) and have made comments in the results (lines 293-300, pages 17) and discussion (line 402-404 page 19) sections. In addition, we will update the PROSPERO protocol accordingly.

Through this additional analysis, as well as a funnel plot, we conclude there is evidence of publication bias however the small numbers mean that some caution is required in the interpretation of these plots.

2) The overall prevalence difference observed could at least be partly attributed to the fact that fibromyalgia is much more prevalent among women (see: Cooksey R. et al. Exploring gender differences, medical history, and treatments used in patients with fibromyalgia in the UK using primary-care data: a retrospective, population-based, cohort study. The Lancet Rheumatology, 2022, 4, S20). The authors are kindly suggested to further discuss this key issue, by additionally implementing any proper data analysis.

Thank you for this helpful comment, since differences in prevalence by chronic pain type are central to this paper. We have refined our argument in the discussion section by citing known differences in prevalence and sex differences in certain pain types, such as fibromyalgia, and have reinforced that there will be heterogeneity in findings depending on how pain is measured. Moreover, our meta-analysis includes sub-analysis by pain type, highlighting the higher prevalence in women in studies of fibromyalgia compared with in studies of general chronic pain. In addition, due to the higher prevalence, the sex difference is greater in these studies too. We have expanded the discussion of this subgroup analysis (lines 339-342, 348-353, page 18). We also feel that differences in findings between generic chronic pain studies and fibromyalgia studies could be attributable to chronic pain measurement. Studies of generic chronic pain included in this review follow the International Association for the Study of Pain definition of pain for over 3 months. This definition does not consider pain impact and severity, primary or secondary pain, and other qualifiers. The studies using this criterion are unable to differentiate between constant and intermittent pain – making cases of generic chronic pain less specific and distant from clinical presentations. These considerations are reflected in lines 360-365, page 18. Since chronic pain prevalence varies across the life course, our results are interesting to those studying chronic pain subtypes (including fibromyalgia) during midlife.

Please find the response to the minor issues:

1) Line 25 (Abstract): The authors are kindly suggested to define the abbreviation "FM".

We have removed the abbreviation FM throughout the text and opted for the long form (fibromyalgia) to avoid confusion.

2) Line 61: The authors are kindly suggested to provide the detailed search algorithm for at least one database.

We have provided this. Edits can be found on line 69, page 3.

3) Line 191 (Figure 3): The authors are kindly suggested to add 95% CI for each study as well as subtotals for each subgroup.

This plot provides estimates for all studies where the RR was available. However, for multiple studies information on counts was not available to calculate the standard error, and therefore the 95% CI, of the RR. It is therefore also not possible to provide the overall estimates as they depend on all studies estimates having a standard error and this is why all these studies are not included in meta-analysis.  Edits can be found in lines 208-212, page 12. In addition, we highlight the need for reporting sex prevalence data in CP literature on line 388-389, page 19.

We hope this satisfies your queries and we remain available for further clarification if necessary.

Kind regard,

Catherine Borra

(on behalf of the authors)

Reviewer 2 Report

Comments and Suggestions for Authors
  1. The search was restricted to four databases; no trial registries, grey literature, or non-English records were sought, risking language and publication bias.
  2. Only “population-representative” designs were eligible, yet no formal quality-of-evidence appraisal (e.g., GRADE) is reported.
  3. Out of 18 studies only eight entered the primary meta-analysis; the authors fail to explain why the remaining ten (with extractable data) were excluded from pooling.
  4.  Critical details—response rates, sampling frames, chronic pain case definitions—are not tabulated, preventing risk-of-bias judgement.
  5. Sensitivity analyses (pain type, definition, region) did not employ meta-regression nor account for multiple testing.
  6. A 2025 comprehensive systematic review of chronic pain in 862 013 European adults already reported female predominance across age strata, including mid-life . The additional value of the current review is therefore marginal.

Author Response

Dear Reviewer 2,

We welcomed your input and would like to thank you for your comments. Here is our response:

1) The search was restricted to four databases; no trial registries, grey literature, or non-English records were sought, risking language and publication bias.

Thank you for your comment. While we decided to assess journal articles from peer reviewed publications to benchmark quality, we are aware that restrictions to our search (particularly our language restriction) may lead to bias. Trial registries were not relevant as they provide observational data and would not provide unbiased prevalence due to selection criteria into trials. Grey literature was not included in the search protocol, although this may be relevant to subsequent reviews. We have acknowledged strengths and limitations of our search within the discussion section. This can be found in lines 394-396, page 19.

2) Only “population-representative” designs were eligible, yet no formal quality-of-evidence appraisal (e.g., GRADE) is reported.

We carefully considered quality-of-evidence appraisal in the design of this systematic review and considered different tools. GRADE (Grading of Recommendations, Assessment, Development and Evaluation) is a tool for rating the strength of evidence and certainty in recommendations for clinical question (Manya Prasad, Introduction to the GRADE tool for rating certainty in evidence and recommendations, Clinical Epidemiology and Global Health, Volume 25, 2024,101484,

ISSN 2213-3984,https://doi.org/10.1016/j.cegh.2023.101484.). As such, we opted for a tool that was specifically designed for quality assessment of prevalence studies. In their systematic review of tools to assess prevalence studies (Celina Borges Migliavaca, Cinara Stein, Verônica Colpani, Zachary Munn, Maicon Falavigna, Quality assessment of prevalence studies: a systematic review, Journal of Clinical Epidemiology, Volume 127, 2020, Pages 59-68, ISSN 0895-4356,

https://doi.org/10.1016/j.jclinepi.2020.06.039) the authors review the risk of bias tool employed in our systematic review (Hoy et al, 2012). We chose this tool as it has been used to assess pain prevalence and is a modification of a previous checklist specifically developed by Leboeuf-Yde and Lauritsen to assess study quality in a review of the prevalence of low back pain. While our tool assesses risk of bias, it relies on a ten item quality assessment including the following questions:

-        Was the study’s target population a close representation of the national population in relation to relevant variables?

-        Was the sampling frame a true or close representation of the target population?

-        Was some form of random selection used to select the sample, OR was a census undertaken?

-        Was the likelihood of nonresponse bias minimal?

-        Were data collected directly from the subjects (as opposed to a proxy)?

-        Was an acceptable case definition used in the study?

-        Was the study instrument that measured the parameter of interest shown to have validity and reliability?

-        Was the same mode of data collection used for all subjects?

-        Was the length of the shortest prevalence period for the parameter of interest appropriate?

-        Were the numerator(s) and denominator(s) for the parameter of interest appropriate?

-         

Items 1 to 4 assess the external validity of the study (domains are selection and nonresponse bias), and items 5 to 10 assess the internal validity (items 5 to 9 assess the domain of measurement bias, and item 10 assesses bias related to the analysis).

We have added a table reporting the risk of bias assessment for all articles within the supplementary materials, table B2. In addition, we have expanded the methods section to include how study quality was assessed as part of the risk of bias assessment in lines 95-107, pages 3-4.

3) Out of 18 studies only eight entered the primary meta-analysis; the authors fail to explain why the remaining ten (with extractable data) were excluded from pooling.

Thank you for flagging this issue. We conducted the meta-analysis on eight studies which provided the necessary data from which sample sizes and counts could be derived – we have listed these in the footnotes to Table 2 in lines 194-198, page 11. The remaining studies reported percentages of participants with CP without providing the relevant sample sizes and/or counts with and without CP  or reported difference in prevalence but did not include counts for participants without CP. We have expanded on this in lines 181-189, page 10, and we have commented on the limitations in the discussion section in lines 386-392, page 19). Please also see the response to point 3 by reviewer 1.

4) Critical details—response rates, sampling frames, chronic pain case definitions—are not tabulated, preventing risk-of-bias judgement.

Thank you for this comment, we have added a table with all these details to the supplementary materials, Table B1, and referenced it in the manuscript in line 146, page 5.

5) Sensitivity analyses (pain type, definition, region) did not employ meta-regression nor account for multiple testing.

We have run the meta-regression and do find a significant difference between estimates for CP and fibromyalgia but not for other subgroup analyses. However, guidance from the Cochrane Collaboration advises that for undertaking simple regression analyses, at least ten observations (i.e. ten studies in a meta-analysis) should be available for each characteristic modelled. Therefore with only 8 studies in the meta-analysis we do not present the results as they will be uninformative seethe section on meta-regression in chapter 10 of the Cochrane Handbook for Systematic Reviews https://www.cochrane.org/authors/handbooks-and-manuals/handbook/current/chapter-10#section-10-11-4). The edits to the manuscript can be found in lines 401-403 page 19.

We acknowledge that we performed multiple subgroup analyses but these were specified in the protocol. This pre-specification reduces the likelihood of spurious findings, by limiting the number of subgroups investigated, and by preventing knowledge of the study results influence the choice of subgroups to be analysed. As these are investigations of heterogeneity and we discuss results in terms of estimates and 95%CIs rather than p-values we do not adjust for multiple testing. Further, given the small number of studies we were able to meta-analyse, we indicate that any subgroup comparisons should be interpreted with caution.  

6) A 2025 comprehensive systematic review of chronic pain in 862 013 European adults already reported female predominance across age strata, including mid-life. The additional value of the current review is therefore marginal.

Thank you for referencing the excellent paper by Rometsch et al (2025). This is an analysis of chronic pain prevalence in literature from four databases. While the review focuses on point prevalence, they do describe estimates for sex and age separately, and within the section pertaining to features associated to 6-12 month prevalence of chronic pain the authors comment on sex and age together citing two studies which present data for both sex and age together. These results are presented narratively. Because the Rometsch et al (2025) systematic review did not aim to derive estimates for chronic pain by age it does not present an in-depth assessment (i.e. meta-analysis) of the topic, and rather confirms the known pooling of chronic pain in later life, and the higher prevalence across the life course in females. We consider that our review adds considerable depth on the topic of sex differences in chronic pain prevalence by providing a meta-analysis of data available globally. We have included reference to the Rometsch et al (2025) study in the introduction in lines 54-56, page 2.

Kind regards,

Catherine Borra

(on behalf of the authors) 

Round 2

Reviewer 1 Report

Comments and Suggestions for Authors I have studied the revised version of the manuscript entitled "Sex differences in the prevalence of chronic pain in mid-life: a systematic review and meta-analysis" by Borra C. et al. The authors have adequately serponded to all queries raised. The manuscript has been improved. There are no additional issues.

Reviewer 2 Report

Comments and Suggestions for Authors

After modification, I agree to the publication of this review.